# Improved Remotely Sensed Total Basin Discharge and Its Seasonal Error Characterization in the Yangtze River Basin

**DOI:** 10.3390/s19153386

**Published:** 2019-08-01

**Authors:** Yutong Chen, Hok Sum Fok, Zhongtian Ma, Robert Tenzer

**Affiliations:** 1School of Geodesy and Geomatics, Wuhan University, Wuhan 430079, China; 2Key Laboratory of Geospace Environment and Geodesy, Ministry of Education, Wuhan University, Wuhan 430079, China; 3Department of Land Surveying and Geo-informatics, The Hong Kong Polytechnic University, Hong Kong 999077, China

**Keywords:** total basin discharge, GRACE satellite gravimetry, TRMM satellite precipitation, MODIS satellite-derived evapotranspiration, Yangtze River Basin

## Abstract

Total basin discharge is a critical component for the understanding of surface water exchange at the land–ocean interface. A continuous decline in the number of global hydrological stations over the past fifteen years has promoted the estimation of total basin discharge using remote sensing. Previous remotely sensed total basin discharge of the Yangtze River basin, expressed in terms of runoff, was estimated via the water balance equation, using a combination of remote sensing and modeled data products of various qualities. Nevertheless, the modeled data products are presented with large uncertainties and the seasonal error characteristics of the remotely sensed total basin discharge have rarely been investigated. In this study, we conducted total basin discharge estimation of the Yangtze River Basin, based purely on remotely sensed data. This estimation considered the period between January 2003 and December 2012 at a monthly temporal scale and was based on precipitation data collected from the Tropical Rainfall Measuring Mission (TRMM) satellite, evapotranspiration data collected from the Moderate Resolution Imaging Spectroradiometer (MODIS) satellite, and terrestrial water storage data collected from the Gravity Recovery and Climate Experiment (GRACE) satellite. A seasonal accuracy assessment was performed to detect poor performances and highlight any deficiencies in the modeled data products derived from the discharge estimation. Comparison of our estimated runoff results based purely on remotely sensed data, and the most accurate results of a previous study against the observed runoff revealed a Pearson correlation coefficient (PCC) of 0.89 and 0.74, and a root-mean-square error (RMSE) of 11.69 mm/month and 14.30 mm/month, respectively. We identified some deficiencies in capturing the maximum and the minimum of runoff rates during both summer and winter, due to an underestimation and overestimation of evapotranspiration, respectively.

## 1. Introduction

Total basin discharge (TBD) is a fundamental water balance component of river basins [1,2] and it has been traditionally measured at in-situ hydrological stations near estuary mouths. TBD can be converted into surface runoff (*R*) when considering a surface area unit over the entire river basin. Continuous TBD time series are necessary for the monitoring of hydrological extremes (i.e., droughts and floods) in deltaic regions. Such monitoring is important for better water management, allowing an increase in water usage efficiency and minimizing unpredictable human, agricultural, and economic losses [3,4,5,6,7]. Nevertheless, a comprehensive global river water discharge (RWD) observing network has not been established yet [8]; moreover, the number of in-situ stations has been decreasing since the late 1970s due to the absence of sufficient funding for the upgrade and maintenance of facilities [9,10]. Consequently, a gauge-independent method, which would provide a synoptic mean of observing the global RWD, remains elusive [11].

Remotely sensed RWD estimations based on the most recent (passive and active) remote sensing (RS) advances, have been demonstrated to be viable alternatives [12]. Over the long period, it is anticipated that remotely sensed RWD estimations will compensate the scarcity of in-situ hydrological data, particularly in remote regions. Current remotely sensed RWD methods can be classified into the following four categories:(1)The correlation between passive remotely sensed variables (e.g., Normalized Difference Vegetation Index (NDVI), and Land Surface Temperature (LST) [13,14,15,16]) and the water level (WL) or RWD data [17,18,19,20,21,22]. Notably, the above remotely sensed variables have no direct causal relationship with RWD.(2)The calculation of the RWD from passive remotely sensed hydraulic variables, via hydraulic geometric equations (e.g., [23,24,25,26,27,28,29]). In this case, the accuracy of RWD estimation would be region-dependent: the resolution of RS images might be insufficient to detect small changes in river width [30] and the roughness coefficient might be unavailable in some regions [31,32].(3)The correlation between active remotely sensed WL (collected from satellite radar altimetry) and in-situ RWD data (collected from hydrological stations) via stage–discharge rating curves [1,33,34,35]. In this case, the altimetry radar signals are partially contaminated by land surfaces when crossing rivers with short widths: the accuracy of the observed altimetric WL is limited (e.g., [36]).(4)The calculation of the RWD near the estuary mouth (or of *R*), based on the land water balance [37] or on the combined land–atmosphere water balance equation [38]. However, both observed and modeled data of various qualities have been utilized in previous studies based on this process (e.g., [39,40]).

The fourth method represents a purely hydrological modeling method that can be applied to the basin-wide estimation of *R* and does not rely on in-situ ground observations. This method involves the combined use of remotely sensed precipitation (*P*), terrestrial water storage (*S*), modeled evapotranspiration (*ET*) and/or modeled atmospheric moisture budget data (i.e., moisture flux divergence (∇·Q), and the changes in the total column water vapor ($\frac{\partial W}{\partial t}$)). Both in the case of land-based (i.e., R=P−ET−ΔS) [41,42] and combined land–atmosphere water balance (i.e., $R=-\Delta S-\frac{\partial W}{\partial t}-\nabla \cdot Q$) e.g., [43] equations, the quality of the modeled *ET* and of the atmospheric moisture budget results is limited by large uncertainties [44,45,46]. This is due to the fact that different land cover types and physical assumptions are considered for their calculation. Moreover, before the launch of Gravity Recovery and Climate Experiment (GRACE), the *S* values were unavailable and their change (ΔS) was assumed to be zero [41,47]. This assumption caused further uncertainties in the estimation of the remotely sensed *R*.

The land–atmosphere water balance equation used for *R* estimation has been applied efficiently for the first time to the Amazon and Mississippi river basins [38]. The authors of that study used GRACE *S* and a modeled atmospheric moisture budget operational forecast analysis (provided by the Environmental Protection-National Center for Atmospheric Research (NCEP-NCAR) [48] and by the European Centre for Medium-Range Forecasts (ECMWF)). The estimated *R* for the Yangtze river basin (YRB) closely agrees with in-situ observed *R*: the peak-to-peak Pearson correlation coefficient (PCC) was equal to 0.92 [44]. Ferreira et al. (2013) [37] estimated *R* for the same river basin but based on the land-based water balance equation (using remotely sensed *P* from the Tropical Rainfall Measuring Mission (TRMM) [49], ΔS from GRACE [50], and modeled *ET* data from the Global Land Data Assimilation System (GLDAS) [45]). The result of Ferreira et al. (2013) [37] indicated a good agreement between the estimated *R* and the in-situ observed *R* time series (i.e., PCC 0.74 and root-mean-square error (RMSE) 14.30 mm/month). 

The *ET* obtained from Moderate Resolution Imaging Spectroradiometer (MODIS) could conveniently replace the modeled *ET*, since the Penman–Monteith equation provided a better representation of croplands and grasslands [51]. These land cover types are densely situated within the YRB, particularly at the middle and lower reaches of the YRB. Previous accuracy assessment did not account for the potential existence of seasonal error characteristics and for deficiencies in the modeled data products (for the estimation of *R)*. This study aims to address such issues; we applied a purely remotely sensed data-driven method based on the land water balance equation and able to estimate *R* in the YRB at a monthly temporal scale. This data-driven method is established using water balance equations in combination with TRMM *P*, MODIS *ET*, and GRACE ΔS data. 

In China, food security relies on the water resources of the YRB; hence, this river basin is one of the most significant study areas [52]. A number of water management projects have been operated for adjustment of *R* during severe drought and flood seasons [53,54]. Other human activities, such as damming, groundwater withdrawal, water consumption, and land use change, can have substantial impacts on *R* of the river basin [55,56]. Land-use change accounts for <0.2% of the change in the runoff trend [57], while damming, groundwater withdrawal, and water consumption account for <2%, <1%, and <10% of the changes in the annual runoff [58], respectively. However, other research studies have indicated climate change as the main factor affecting *R* (accounting for 90% of the change in *R* in the YRB) [57,59,60,61]. Additional studies have indicated that human activities in the YRB might significantly influence changes in the sediments, but not in *R* [60,62].

Notably, the *R* values estimated from the water balance equation were calculated by subtraction among the remotely sensed *P*, *ET*, and ∆*S*: the systematic effect caused by the YRB environment and by human activities should have been partly mitigated by this subtraction process. The aforementioned information partially supports our data-driven *R* estimation for the YRB, based purely on the above remotely sensed hydrological variables on the application of the water balance equation. After calculating the resulting *R*, we compared our results with those published in Syed et al. (2009) [44] and Ferreira et al. (2013) [37]; moreover, we performed a seasonal accuracy assessment of the observed *R* time series, in order to evaluate the performance of our method during all seasons. Finally, we discussed the deficiencies of modeled data products and tried to explain the reasons for different seasonal discrepancies between the *R* obtained through our method and the observed in-situ data.

This paper is organized as follows: Section 2 presents the geographic environment of the YRB; Section 3 presents the methodology, data used in this study, and their validation; Section 4 demonstrates the validity of the estimated *R* and compares our results with those of previously published studies; and Section 5 summarizes our conclusions.

## 2. Geography of the Yangtze River Basin (YRB)

Situated between 25°–35° N and 91°–122° E, within the subtropical zone, the Yangtze River is the world’s third longest river and the first in China [63]. It flows from the three-rivers headwater region (northeastern Tibetan Plateau) to the East China Sea, for a total length of 6300 km and with a total drainage basin area of 1,800,000 km^2^ [64,65]. The whole YRB (Figure 1) can be divided into three reaches, based on its stepped topography and on the river channel cross-sections: an upper (from the Tibetan Plateau to Yichang), a middle (from Yichang to Hukou), and a lower reach (from Hukou to the river estuary) [66].

The majority of the YRB is subject to the subtropical monsoons, in particular the Indian Summer Monsoon (ISM) and the East Asian Summer Monsoon (EASM) [67,68]. Generally, the surface air temperature in the YRB reaches maxima in summer and minima in winter [69]; moreover, *P* concentrates over the middle-lower reach of the YRB. The mean annual *P* rate in correspondence of the upper reach (i.e., 270–500 mm) is significantly lower than that in correspondence of the middle-lower reach (i.e., 1600–1900 mm) [70]. Controlled by the monsoonal climate, the rainy season starts from April and ends in September, peaking in July every year [71].

## 3. Methodology, Data Description and Validation

### 3.1. Methodology

The terrestrial water balance equation, based on the principle of mass conservation, has been used to estimate the remotely sensed runoff *R* for the YRB [41,42]:(1)R=P−ET−ΔS where *P* is the precipitation (mm/month), *ET* the evapotranspiration (mm/month), ∆*S* the terrestrial water storage change (mm/month), which can be expressed as:(2)$$\Delta S=\frac{1}{2}\left(S_{i+1}-S_{i-1}\right)$$ where $S_{i}$ is water storage anomaly (mm/month) of month *i*.

Table 1 summarizes all the remotely sensed data products and the in-situ gauge observations used in this study. All the in-situ gauge data time series were obtained from the Changjiang Water Resources Commission (Ministry of Water Resources; http://www.cjh.com.cn). Notably, the remotely sensed *P* data from the TRMM, the *ET* data from the MODIS, and the *S* data from GRACE have different spatial resolutions (Table 1) and need to be unified. Therefore, the TRMM and MODIS data were resampled at a 1°×1° resolution, whereas the GRACE data were interpolated at a 1°×1° resolution. The remotely sensed data were then validated against the in-situ gauge at point locations. No in-situ *ET* gauge stations were available in the YRB [51]; hence, we needed to use the evaporation (*E*) in-situ measurements to infer the *ET*. Since terrestrial water storage measurements were unavailable at a point location, the GLDAS was employed to validate the data in this study.

Once the steps described above were completed, all the *P*, *ET*, and ΔS values gridded within the YRB up to Datong station location were individually summed up and used to calculate the monthly *R* values between 2003 and 2012 (see Equation (1) and Figure 1). To be comparable with the unit of the estimated *R* values, the daily observed discharge ($m^{3}/$sec) data of Datong station were combined to in the form of monthly discharge values and subsequently divided by the approximate total drainage area, up to Datong station (i.e., $1.7\times 10^{6}\;{\mathrm{km}}^{2}$) to obtain the monthly *R* per unit area (mm/month) (Figure 1). 

### 3.2. In-Situ Hydrological Gauge Stations as Reference Data

In proximity of the estuary mouth, the river channel is seasonally affected by a combination of fluvial (i.e., discharge) and marine (i.e., tide and wave) processes. The Datong station was chosen to validate the remotely sensed *R* (Figure 1), since it is sufficiently far from the estuary. For consistency with the RS data, we used the RWD time series collected at the Datong station between January 2003 and December 2012. The observed RWD data (m^3^/s) were converted into monthly *R* rates (mm/month) per unit area of the YRB. To validate the remotely sensed *P* (TRMM) and *ET* (MODIS) data, we considered the observed *P* and the inferred *ET* using Equation (6) of Zhang et al. (2001) [72] from the observed *E* data time series collected at stations along the upper (i.e., Pingshan), middle (i.e., Chenglingji), and lower reaches (i.e., Datong) of the Yangtze River (Figure 1).

The Datong station time series showed an annual peak and trough in July and February, respectively (Figure 2). The exceptionally large *R* that occurred in July 2010 was likely caused by a moderate El Niño event that started in autumn 2009, as also indicated by the Oceanic Niño Index (https://ggweather.com/enso/oni.htm).

### 3.3. Remotely Sensed Hydrological Variables and Their Validation

#### 3.3.1. Evaluation Metrics

To estimate the remotely sensed *R* for the YRB, several remotely sensed datasets have been considered (i.e., *P* (TRMM), *ET* (MODIS), and *S* (GRACE) data). These datasets and the estimated *R* were subjected to an accuracy evaluation, by comparing them to the gauge station observed time series. In particular, the remotely sensed data were compared to the gauge station observed time series using the PCC and the RMSE, whereas the *R* values estimated from the water balance model were compared to the gauge station observed time series using the PCC, RMSE, and the Nash–Sutcliffe model efficiency (NSE) coefficient. The PCC is a number that represents the strength of the linear relationship between two data time series; it can be calculated as follows:(3)$$\mathrm{PCC}=\frac{\sum_{i=1}^{N}\left(X_{o}^{i}-\bar{X_{o}}\right)\left(X_{m}^{i}-\bar{X_{m}}\right)}{\sqrt{\sum_{i=1}^{N}{\left(X_{o}^{i}-\bar{X_{o}}\right)}^{2}}\sqrt{\sum_{i=1}^{N}{\left(X_{m}^{i}-\bar{X_{m}}\right)}^{2}}}$$

The RMSE represents the accuracy of the estimation; It can be calculated as follows:(4)$$\mathrm{RMSE}=\sqrt{\frac{\sum_{i=1}^{N}{\left(X_{m}^{i}-X_{o}^{i}\right)}^{2}}{N}}$$

The NSE coefficient proposed by Nash and Sutcliffe (1970) [73] is usually used to assess hydrological models; it can be calculated as follows:
(5)$$\mathrm{NSE}=1-\frac{\sum_{i=1}^{N}{\left(X_{m}^{i}-X_{o}^{i}\right)}^{2}}{\sum_{i=1}^{N}{\left(X_{o}^{i}-\bar{X_{o}}\right)}^{2}}$$ where $X_{o}$ and $\bar{X_{o}}$ are the gauge observed and the average gauge observed values, respectively; $X_{m}$ and $\bar{X_{m}}$ represent the remotely sensed *R* and the average remotely sensed values, respectively; and *N* corresponds to the number of observations within the time series. The closer the NSE is to 1, the better the performance of $X_{m}$ is. Notably the numerical value of NSE should be equivalent to the coefficient of determination (R^2^), a statistical measure used to predict future outcomes and test hypotheses.

#### 3.3.2. Remotely Sensed Precipitation

The Tropical Rainfall Measuring Mission (TRMM) is a satellite mission responsible for the spatiotemporal measurement of *P* over a large latitude interval (between 50° S and 50° N [74]). The monthly gridded global *P* data (TRMM3B43 V7) used in this study, having a spatial resolution of 0.25° [50], were obtained from the Goddard Earth Sciences Data and Information Services Center (GES DISC) managed by NASA (https://disc.gsfc.nasa.gov/datasets/TRMM_3B43_V7/summary downloaded on 14 June 2018). Notably the TRMM 3B43 V7 data products are not purely a satellite-based; in fact, the Global Precipitation Climatology Centre (GPCC) gauge data were used for bias correction and calibration [75,76,77]. 

Three hydrological stations were selected in the YRB and considered for the comparison between the satellite *P* (TRMM) and the observed monthly *P* data. The result showed that the satellite *P* (TRMM) and the observed *P* data were strongly correlated with each other in the case of all three stations, their the PCCs were equal to 0.82 (Pingshan station), 0.95 (Chenglingji station), and 0.88 (Datong station), respectively (Table 2). Moreover, their respective RMSE values indicated similar conditions along the upper, middle, and lower river reaches, although the highest correlation was identified in correspondence of the middle reaches. Overall, the TRMM and observed *P* data series of the YRB were in good agreement with each other; hence, they were considered reliable for our study (Figure 3).

#### 3.3.3. Remotely Sensed Evapotranspiration

Evapotranspiration (*ET*) is a water vapor that re-enters the atmosphere. It includes the evaporation from soil moisture and the transpiration from land vegetation [78]. The MOD16A2 *ET* data product was prepared based on the MODIS and Global Meteorological Assimilation Office (GMAO) data products using the Penman–Monteith equation [79], having the spatial coverage of 80° N–60° S and 0° E–360° E. These data products were obtained from the MODIS Global Evaporation Project (i.e., MOD16A2) dataset, managed by the Land Processes Distributed Active Archive Center (LP DAAC) of NASA (https://lpdaac.usgs.gov/dataset_discovery/modis/modis_products_table). These data products were also available at the Numerical Terradynamic Simulation Group in the University of Montana (http://www.ntsg.umt.edu/project/modis/).

Some controversy exists regarding the uncertainty of regional or global-scale *ET* [80,81]. Therefore, it was necessary to compare the MODIS *ET* data to the inferred *ET* using Equation (6) of Zhang et al. (2001) [72] from the observed *E* data. The results of this comparison showed that the MODIS *ET* were strongly correlated with the inferred *ET* along the middle and lower river reaches: the PCCs of 0.87 (Pingshan station), 0.71 (Chenglingji station), and 0.80 (Datong station), respectively (Table 3). Meanwhile, the respective magnitude of the RMSE values was apparently dependent on the amount of *ET* (Figure 4). 

The MODIS *ET* data were also compared to the *ET* modeled data (derived from the GLDAS). Four different land surface models (i.e., the common land model (CLM), the Mosaic, the National centers for environmental prediction/Oregon State University/Air force/Hydrologic research lab (NOAH), and the variable infiltration capacity (VIC) model) were extracted. The MODIS *ET* and the four sets of GLDAS modeled data were characterized by strong and consistent PCCs; however, the RMSE values revealed diverse performances (Table 4). These results can be attributed to a consistency in the seasonal patterns, combined with a considerable variation in the occurrence of maxima during the summer season (Figure 5). For the entire YRB, the CLM *ET* data showed the best match with the MODIS *ET* data.

#### 3.3.4. Remotely Sensed Terrestrial Water Storage

GRACE is a satellite mission responsible for measuring Earth’s time-variable gravity changes, which allow inference of global *S* variations (in terms of Equivalent Water Height (EWH)). The EWH were computed by the Center for Space Research (CSR), GeoForschungsZentrum (GFZ), the Institute of Theoretical geodesy and Satellite Geodesy (ITSG), and the Jet Propulsion Laboratory (JPL) GRACE Level-2 Release 05 (RL05) GSM monthly gravity fields for the period between January 2003 and December 2012 in the form of spherical harmonic coefficients (SHCs) [82].

Unified postprocessing procedures were applied to the SHCs before converting them into *S* data. The degree-one terms were restored to correct for the geocenter motion [83], while the term $C_{20}$ in the form of the SHCs was replaced by the Satellite Laser Ranging (SLR) results [84]. A destriping process and a Gaussian filtering (radius 350 km) were applied in order to reduce the spatially correlated errors of the *S* [85,86]. Afterward, the values of *S* time series were calculated considering a regular grid using Equation (14) in [87] and dividing the result by water density. Additionally, the sum of the preceding and succeeding two-month data were averaged in order to compensate for (i.e., interpolating) the missing monthly *S* data.

There is controversy about the uncertainty of the GLDAS-modeled *S* data, especially when compared to remotely sensed GRACE *S* data (e.g., [88]). Hence, the GRACE *S* data needed to be validated by comparing them with the GLDAS-modeled *S* data. Only the GRACE *S* data obtained from the CSR are shown in Table 5, since all GRACE data products exhibited similar patterns (independently from the position of the correspondent stations). We noted a large disparity and a low correlation between the GRACE and the GLDAS-modeled *S* data (Table 5), particularly in correspondence of the river upper reaches. Moreover, the GLDAS-modeled *S* data series obtained from different sources were considerably different among each other (Figure 6). In fact, while the modeled CLM *S* data displayed small variability, those from Mosaic displayed large fluctuations. The modeled *S* data from the VIC and the NOAH showed modest fluctuation patterns, similar to that of the GRACE data. Furthermore, leakage errors in the GRACE *S* data and the uncertainties inherent to the GLDAS modeled *S* data could have contributed to their poor correlation (e.g., [89]). Therefore, it is difficult to assess the accuracy of the GRACE *S* simply based on the GLDAS modeled *S* data: the pure remotely sensed *R* and modeled (i.e., *ET* and/or *S*) datasets need to be further compared with each other. 

## 4. Discussion

The resulting *R* time series based on our purely satellite data-driven method (i.e., MODIS *ET* and several GRACE *S*) were compared to remotely sensed and modeled data products (i.e., ITSG GRACE *S* and several GLDAS *ET*, or MODIS *ET* and several GLDAS *S*), which were based on the same TRMM *P* data (Table 6). Our analysis showed that the MODIS and ITSG, ITSG and CLM, and MODIS and Mosaic were the best data combinations to describe the remotely sensed *ET* and *S* changes (Figure 7). The combination (i.e., ITSG GRACE *S* and CLM *ET*) yields a PCC of 0.73 and an RMSE of 15.69 mm/month, comparable to those reported by Ferreira et al. (2013) [37] (i.e., 0.74 and 14.30 mm/month, respectively). Our metrics were slightly less accurate than those in [37], probably due to the additional three-year time span (between 2010 and 2012) considered in the study: during this period, the occurrence of a moderately strong El Niño event might have created anomalous conditions. In the case of the MODIS *ET* and GRACE *S* data, our data-driven method resulted in a PCC of 0.88 and an RMSE of 11.69 mm/month, indicating a higher accuracy than that obtained in [37]. According to the results, the MODIS16A2 *ET* data are more reliable than those of the GLDAS, at least for the YRB. Probably, this was due to the use of the Penman–Monteith equation during the production of the MODIS16A2 *ET* data: this equation favors the detection of cropland and grassland covers [51], particularly in the YRB.

Our results were validated against those result of Syed et al. (2009) [44] by considering the peak-to-peak correlation between the remotely sensed and the observed *R* (Table 7). We found that all the peak-to-peak correlations were larger than 0.92: our pure data-driven method resulted in a correlation of 0.96, while the methods used in [44] and [37] resulted in correlation coefficients of 0.92 and 0.93, respectively. Overall, the results we derived from TRMM *P*, MODIS *ET* and GRACE *S* data were substantially more accurate than those presented in the previous publications.

The Taylor’s diagram provides a direct way to determine the degree of correspondence between observed and estimated data [90]: it shows the correlation, the RMSE difference, and the standard deviation between two types of data. Figure 8 displayed that four different remotely sensed *R* derived through the purely data-driven method (triangles) overlapped each other: this indicates that the choice of different GRACE *S* in combination with the MODIS *ET* data has a negligible effect on the estimated *R*. However, the results derived from the MODIS *ET* and GLDAS *S*, and GLDAS *ET* and GRACE *S* datasets, showed poor consistency, in particular the latter one. These results highlight that the *ET* is likely the most significant error source during the estimation of the remotely sensed *R*. In summary, the remotely sensed *R* derived from the purely data-driven method is the closest to the observed *R*.

The discrepancy between the observed and evaluated runoff was examined in detail by computing the mean monthly *R* derived from the three dataset combinations. The results derived from the purely data-driven method best matched those of the observations: the estimated *R* values for different seasons match those of the observations with various degrees of accuracies. Our results indicated a maximum and a minimum *R* in July and January, respectively (Figure 9); however, the estimated minimum *R* was inconsistent with that observed *R*, which occurred in February. Apparently, the estimated *R* was underestimated in winter and overestimated in summer (Figure 9a), likely due to the characteristics of the vegetation cover types, which can influence the calculated MODIS *ET* in different ways during dry and humid seasons.

In case of the GLDAS *ET* and ITSG GRACE *S* combination (Figure 9b), the estimated runoff was underestimated between January and August: the GLDAS *ET* data were overestimated (Figure 5), likely due to the effect of incoming radiations and temperature, as stated in [51]. The *S* data derived from the GLDAS land surface model were even less accurate than those described above (Figure 9c): the resulting estimated *R* displayed its maximum one month earlier than the observed time series, causing an underestimation of the *R* between July and December.

In general, the correlation coefficients between the estimated and the observed *R* during spring and autumn were higher than during summer and winter from the three combinations of remotely sensed data (Table 8). These results further confirm our observations based on Figure 9: overall, the rising and falling trends were well captured by our estimated *R*. The higher correlation coefficients during the spring and autumn months probably derived from an overestimation of the modeled *ET* and, possibly, by the low temporal resolution of the data at the monthly scale.

## 5. Conclusions

Previous studies have calculated the time series of monthly TBD (in terms of *R*) in the YRB by applying the water balance equation to a combination of RS and modeled data products. Here, we applied a data-driven method purely based on RS data. Before the investigation, the remotely sensed data were first validated against the in-situ gauge measurements or the inferred measurements at point locations. This validation process indicated that large uncertainties existed in the modeled data products, as verified when the modeled data products were compared with the observed hydrological data collected from the in situ stations or the inferred data, or when the estimated runoff were compared against the observed runoff.

Our best *R* (obtained from purely remotely sensed data) and those of Ferreira et al. (2013) [37] against the observed runoff reveal the PCC of 0.89 and 0.74, and the RMSE of 11.69 mm/month and 14.30 mm/month, respectively: our method showed statistically better results. The peak-to-peak correlation values were also calculated: in this sense, our method produced slightly better results than those of Syed et al. (2009) [44] and Ferreira et al. (2013) [37].

Seasonal error characterization was conducted to assess the performance of our method during specific seasons. We found that the remotely sensed TBD did not accurately capture the maximum and minimum runoff values in summer and winter, respectively. This poor performance could be attributed to an overestimation and underestimation of *ET*, respectively, which depend on the input variables (e.g., vegetation cover types, solar radiation, and temperature) [51]. The *ET* values tend to be overestimated or underestimated when the input variables are instable, particularly in dry and humid seasons. Notably, the low temporal resolution of the data at a monthly scale could have also contributed to these effects. This finding has not been reported in previous studies.

Satellite data products with higher temporal resolution are gradually becoming available (e.g., daily TRMM precipitation [91], eight-day MODIS evapotranspiration [92], and daily GRACE terrestrial water storage data products [93]). Future research might contemplate the application of our proposed method to these new data, while caution has to be taken with the data validation, data postprocessing steps, and the geographic region.

## Figures and Tables

**Figure 1 sensors-19-03386-f001:**
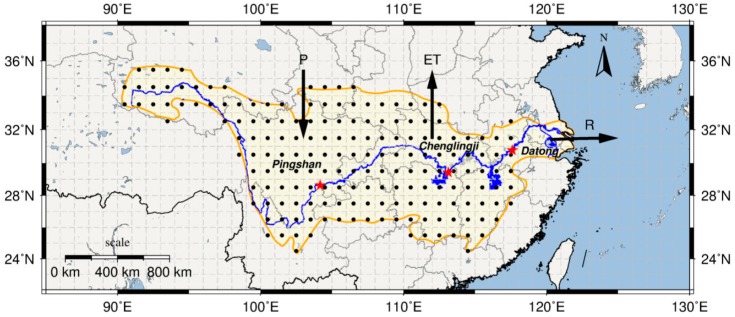
Map of the Yangtze River basin (YRB). Selected hydrological stations (red stars), which collected precipitation, evaporation, and discharge data time series, are situated along the upper, middle, and lower reaches of the Yangtze River. The point locations (black dots) correspond to grid values included in the estimated basin-averaged runoff.

**Figure 2 sensors-19-03386-f002:**
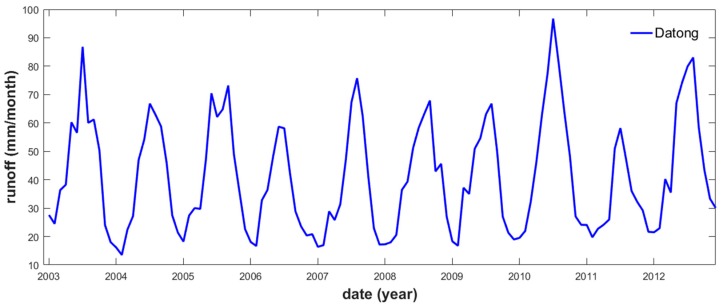
Monthly runoff time series, averaged from the daily discharge data collected at the Datong station (and divided by the basin area).

**Figure 3 sensors-19-03386-f003:**
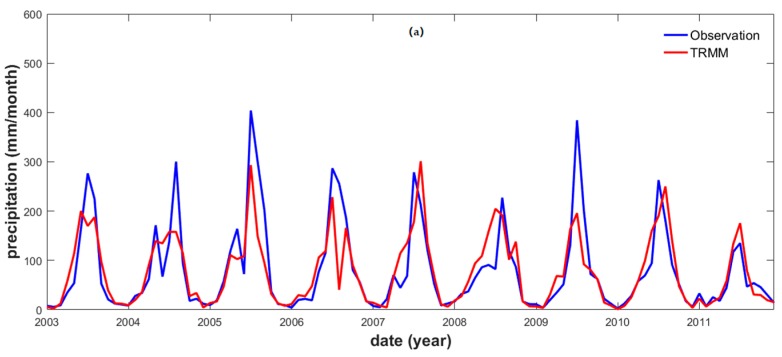
TRMM and observation precipitation time series in correspondence of the (**a**) Pingshan (upper reaches), (**b**) Chenglingji (middle reaches), and (**c**) Datong (lower reaches) stations.

**Figure 4 sensors-19-03386-f004:**
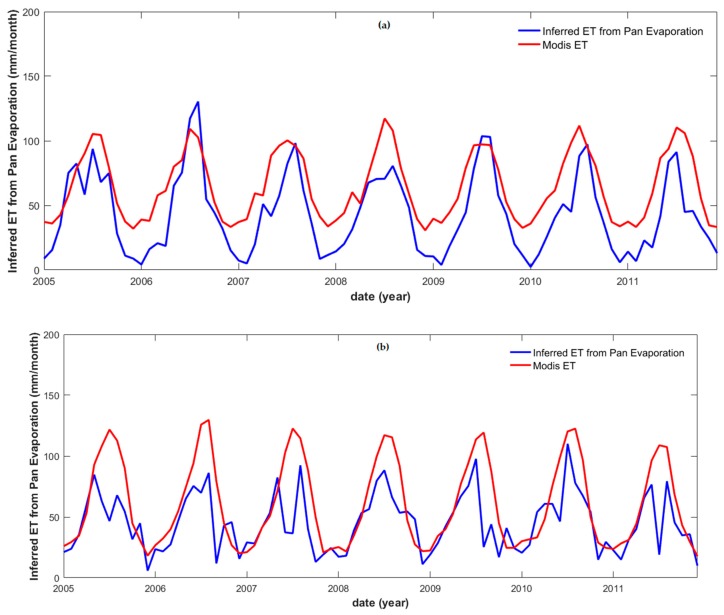
MODIS evapotranspiration and inferred evapotranspiration from observed evaporation data time series in correspondence of (**a**) Pingshan (upper reaches), (**b**) Chenglingji (middle reaches), and (**c**) Datong (lower reaches) stations.

**Figure 5 sensors-19-03386-f005:**
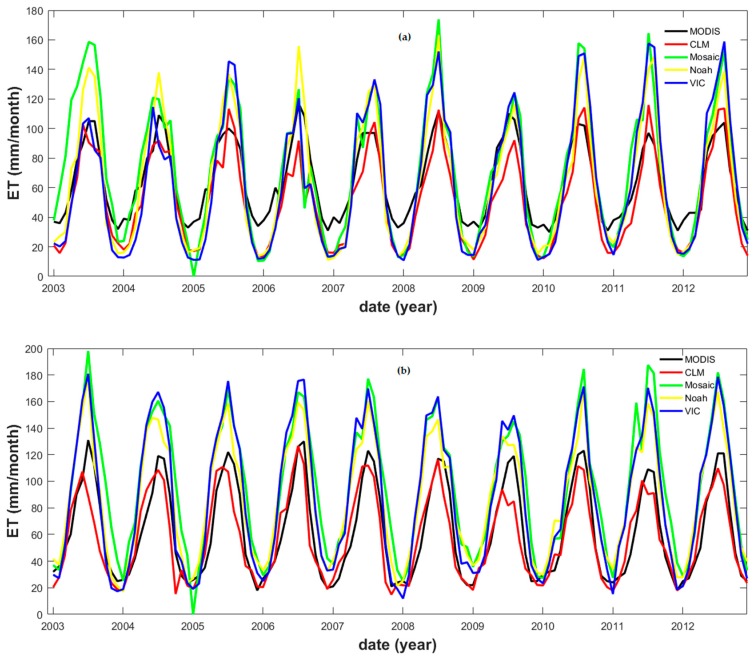
MODIS and GLDAS modeled evapotranspiration time series in correspondence of (**a**) Pingshan (upper reaches), (**b**) Chenglingji (middle reaches), and (**c**) Datong (lower reaches) stations.

**Figure 6 sensors-19-03386-f006:**
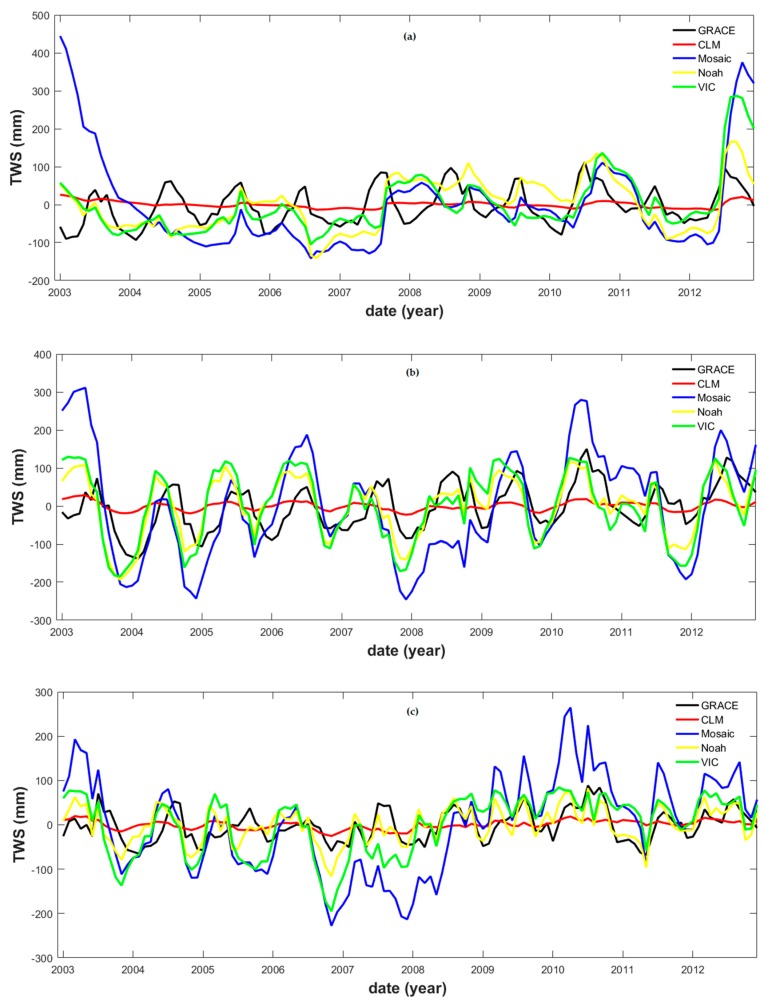
GRACE terrestrial water storage and the GLDAS-modeled time series in correspondence of (**a**) Pingshan (upper reaches), (**b**) Chenglingji (middle reaches), and (**c**) Datong (lower reaches) stations.

**Figure 7 sensors-19-03386-f007:**
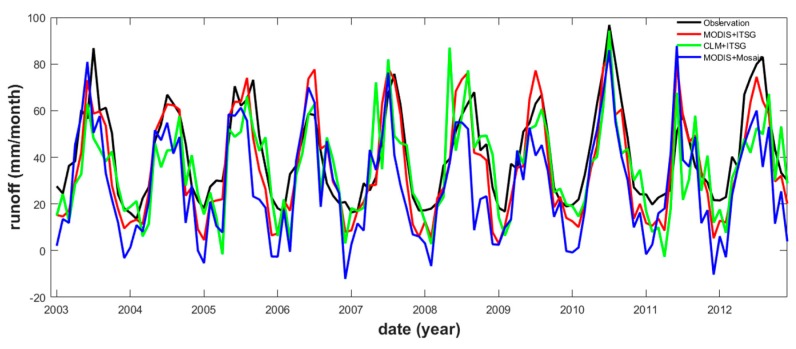
Remotely sensed runoff for the Yangtze river basin derived from the MODIS evapotranspiration and ITSG GRACE terrestrial water storage, CLM evapotranspiration and ITSG GRACE terrestrial water storage, and MODIS evapotranspiration and Mosaic terrestrial water storage datasets.

**Figure 8 sensors-19-03386-f008:**
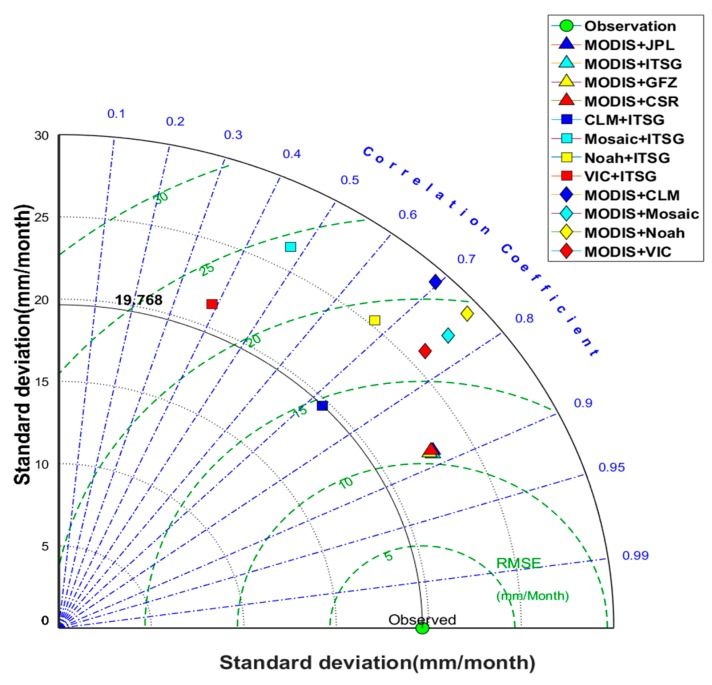
Taylor’s diagram showing the statistics results obtained from the comparison between the observed (green circle) and remotely sensed runoff in the Yangtze River Basin (derived from the MODIS evapotranspiration and several GRACE terrestrial water storage (triangles), several GLDAS evapotranspiration and ITSG GRACE terrestrial water storage (squares), and the MODIS evapotranspiration and several GLDAS water storage (diamonds) data.

**Figure 9 sensors-19-03386-f009:**
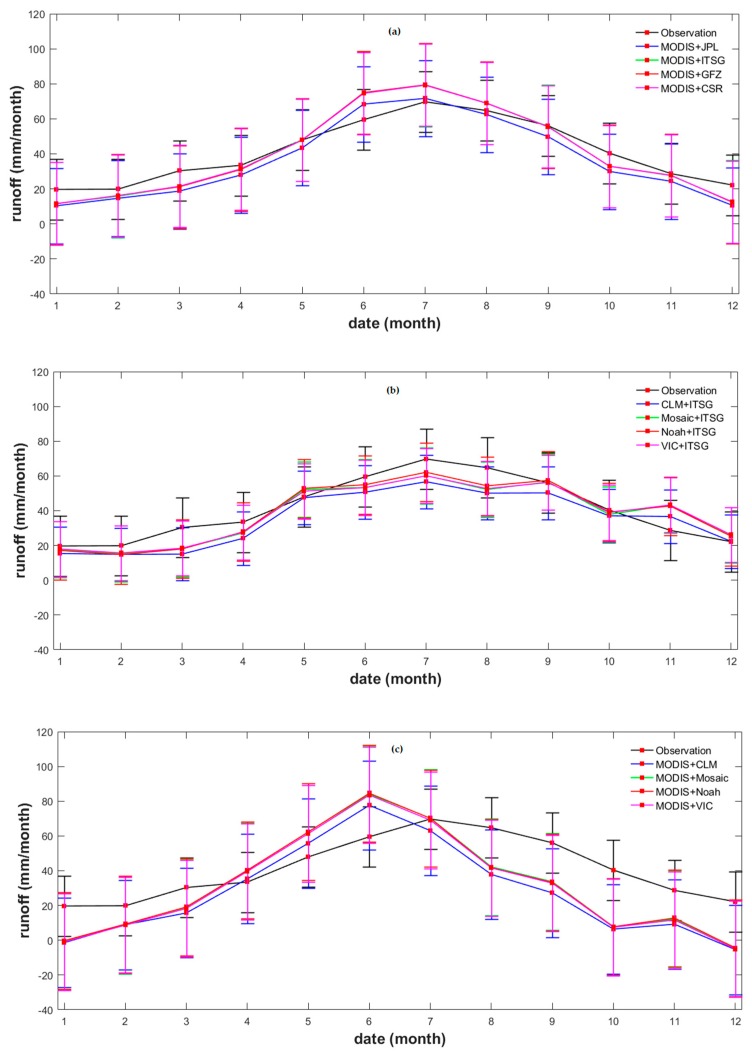
Mean monthly runoff in the Yangtze river basin derived from (**a**) MODIS evapotranspiration and several GRACE terrestrial water storage; (**b**) several GLDAS evapotranspiration and ITSG GRACE terrestrial water storage; and (**c**) MODIS evapotranspiration and several GLDAS terrestrial water storage data.

**Table 1 sensors-19-03386-t001:** Remotely sensed data products and in-situ gauge observations used in this study.

Hydrological Component	Data Source	Spatial Resolution	Temporal Resolution
Precipitation	TRMM 3B43 v7	0.25°×0.25°	Monthly
	Gauges	Point	Daily
Evapotranspiration	MOD16A2	0.5°×0.5°	Monthly
	GLDAS	1°×1°	Monthly
Evaporation	Gauges	Point	Daily
Terrestrial water storage change	GRACE	3°×3°	Monthly
	GLDAS	1°×1°	Monthly
Runoff	Gauges	Point	Daily

**Table 2 sensors-19-03386-t002:** Statistical comparison between the remotely sensed precipitation data and the observed precipitation data at the three selected stations, located along the upper, middle, and lower reaches of the Yangtze River basin.

Station	PCC	RMSE (mm/month)
Pingshan	0.82	50.8
Chenglingji	0.95	32.1
Datong	0.88	52.3

**Table 3 sensors-19-03386-t003:** Statistical comparison between the remotely sensed MODIS evapotranspiration data and the inferred evapotranspiration from the observed evaporation data at the three selected stations along the upper, middle, and lower reaches of the Yangtze River basin.

Station	PCC	RMSE (mm/month)
Pingshan	0.87	26.4
Chenglingji	0.71	29.7
Datong	0.80	29.4

**Table 4 sensors-19-03386-t004:** Statistical comparison between remotely sensed MODIS evapotranspiration and the GLDAS modeled evapotranspiration data at the three selected stations along the upper, middle, and lower reaches of the Yangtze River basin.

Station	GLDAS	PCC	RMSE (mm/month)
Pingshan	CLM	0.94	16.5
	Mosaic	0.89	25.1
	Noah	0.95	19.6
	VIC	0.91	23.3
Chenglingji	CLM	0.92	14.5
	Mosaic	0.93	39.1
	Noah	0.94	29.4
	VIC	0.95	34.0
Datong	CLM	0.96	10.4
	Mosaic	0.91	30.0
	Noah	0.89	22.2
	VIC	0.95	25.5

**Table 5 sensors-19-03386-t005:** Statistical comparison between the remotely sensed GRACE (CSR) and the GLDAS-modeled terrestrial water storage datasets at the three selected stations along the upper, middle, and lower reaches of the Yangtze River basin.

	GLDAS	PCC	RMSE (mm/month)
Pingshan	CLM	0.03	49.09
	Mosaic	0.05	129.50
	Noah	0.38	66.40
	VIC	0.32	75.66
Chenglingji	CLM	0.43	56.93
	Mosaic	0.57	115.04
	Noah	0.52	68.12
	VIC	0.44	84.11
Datong	CLM	0.29	34.17
	Mosaic	0.56	91.10
	Noah	0.56	34.20
	VIC	0.41	60.31

**Table 6 sensors-19-03386-t006:** Statistical comparison between the remotely sensed monthly runoff data derived from the MODIS evapotranspiration and several GRACE terrestrial water storage, several GLDAS evapotranspiration and ITSG GRACE water storage, and MODIS evapotranspiration and several GLDAS terrestrial water storage data, and the ground-based observed runoff.

Dataset Combinations	PCC	RMSE(mm/month)	NSE
MODIS + JPL	0.88	11.89	0.631
MODIS + ITSG	0.89	11.69	0.643
MODIS + GFZ	0.88	11.78	0.638
MODIS + CSR	0.88	11.90	0.630
ITSG GRACE + CLM	0.73	15.69	0.357
ITSG GRACE + Mosaic	0.48	29.26	−1.235
ITSG GRACE + Noah	0.67	20.66	−1.146
ITSG GRACE + VIC	0.39	26.29	−0.805
MODIS + CLM	0.70	24.96	−0.627
MODIS + Mosaic	0.76	19.16	0.004
MODIS + Noah	0.76	20.82	−0.132
MODIS + VIC	0.76	21.41	−0.196

**Table 7 sensors-19-03386-t007:** Peak-to-peak correlation between the remotely sensed runoff and the ground-based observed runoff.

Peak to Peak Correlation	
MODIS + JPL	0.96
MODIS + ITSG	0.96
MODIS + GFZ	0.96
MODIS + CSR	0.96
ITSG GRACE + CLM	0.95
ITSG GRACE + Mosaic	0.94
ITSG GRACE + Noah	0.95
ITSG GRACE + VIC	0.92
MODIS + CLM	0.96
MODIS + Mosaic	0.95
MODIS + Noah	0.97
MODIS + VIC	0.96
Syed et al. (2009) [44]	0.92
Ferreira et al. (2013) [37]	0.93

**Table 8 sensors-19-03386-t008:** Seasonal statistical assessment of the remotely sensed runoff derived from the comparison between the MODIS evapotranspiration and ITSG GRACE terrestrial water storage (MODIS + ITSG), CLM evapotranspiration and ITSG GRACE terrestrial water storage (CLM + ITSG), and MODIS evapotranspiration and Mosaic terrestrial water storage (MODIS + Mosaic), and the ground-based observed runoff data registered at the Datong station.

	Season	PCC	RMSE	NSE
MODIS + ITSG	Spring	0.76	11.00	0.128
	Summer	0.34	13.21	−0.112
	Autumn	0.76	11.88	0.384
	Winter	0.33	10.50	−6.082
CLM + ITSG	Spring	0.50	19.33	−1.693
	Summer	0.46	19.29	−1.37
	Autumn	0.54	13.13	0.248
	Winter	0.44	8.19	−3.31
MODIS + Mosaic	Spring	0.64	14.61	−0.539
	Summer	0.24	19.77	−1.489
	Autumn	0.29	23.64	−1.441
	Winter	0.06	19.60	−23.654

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
