# Peer review of "Improved Remotely Sensed Total Basin Discharge and Its Seasonal Error Characterization in the Yangtze River Basin"

_sensors, 2019, doi:10.3390/s19153386_

Round 1

Reviewer 1 Report

1) This paper used a lot of data, however, many of them are not clearly decribed. For example, the observed ET was used to validate the remotely sensed evapotranspiration, but the author didn't explain how to collecte the observed ET. Therefore, I strongly suggest that the author rewrite the 3. Data Description and Validation Part and add some details.

2) The fatal flaw of this paper is that the author didin't decribe the method of how to use the remotely-sensed data to estimate the total basin discharge. I guess that the authors estimate the total basin discharge based on the water balance. The authors should clearly decribe this issue in the Method Part.

3) I still worry about the accuracy of the TRMM percipitation and GRACE data. The daily TRMM precipitation is very poor and the montly TRMM precipitaiton is also not good. The spatial resolution of GRACE is low, and the accuracy maybe not good. 

4) In China, the runoff is seriously affected by human activities and water management projects, but this paper didn't consider the above factors. So, I think the accuracy of this research is questioned.

5) Language should be improved and polished.

Author Response

Please refer to the attached PDF, as the equations and symbols and tables cannot be displayed. Thank you.

1) This paper used a lot of data, however, many of them are not clearly decribed. For example, the observed ET was used to validate the remotely sensed evapotranspiration, but the author didn't explain how to collecte the observed ET. Therefore, I strongly suggest that the author rewrite the 3. Data Description and Validation Part and add some details.

Response: Please refer to our response to your below question (2).

2) The fatal flaw of this paper is that the author didin't decribe the method of how to use the remotely-sensed data to estimate the total basin discharge. I guess that the authors estimate the total basin discharge based on the water balance. The authors should clearly decribe this issue in the Method Part.

Response: To accommodate your request, we added the methodology description in the beginning of section 3 of the revised manuscript as below:

"3. Methodology, Data Description and Validation

3.1 Methodology

The terrestrial water balance equation, based on the principle of mass conservation, has been used to estimate the remotely-sensed runoff R for the YRB [41-42]:

(1) where P is the precipitation (mm/month), ET the evapotranspiration (mm/month), ∆S the terrestrial water storage change (mm/month), which can be expressed as:

(2) where  is water storage anomaly (mm/month) of month i.

Table 1 summarizes all the remotely-sensed data products and the in-situ gauge observations used in this study. All the in-situ gauge data time series were obtained from the Changjiang Water Resources Commission (Ministry of Water Resources; http://www.cjh.com.cn). Notably, the remotely-sensed P data from the TRMM, the ET data from the MODIS, and the S data from GRACE have different spatial resolutions (Table 1) and need has to be unified. Therefore, the TRMM and MODIS data were re-sampled at a  resolution, whereas the GRACE data were interpolated at a  resolution. The remotely-sensed data were then validated against the in-situ gauge at point locations. No in-situ ET gauge stations were available in the YRB [52]; hence, we needed to use the evaporation (E) in-situ measurements. Since terrestrial water storage measurements were unavailable at a point location, the GLDAS was employed to validate the data in this study.

Once the steps described above were completed, all the P, ET, and  values gridded within the YRB up to Datong station location were individually summed up and used to calculate the monthly R values between 2003 and 2012 (see equation (1) and Figure 1). To be comparable with the unit of the estimated R values, the daily observed discharge (ec) data of Datong station were combined to in the form of monthly discharge values and subsequently divided by the approximate total drainage area, up to Datong station (i.e., ) to obtain the monthly R per unit area (mm/month) (Figure 1)."

Table 1. Remotely-sensed data products and in-situ gauge observations used in this study.

Hydrological   Component

Data Source

Spatial   Resolution

Temporal   Resolution

Precipitation

TRMM 3B43 v7

Monthly

Gauges

Point

Daily

Evapotranspiration

MOD16A2

Monthly

GLDAS

Monthly

Evaporation

Gauges

Point

Daily

Terrestrial   water storage change

GRACE

Monthly

GLDAS

Monthly

Runoff

Gauges

Point

Daily

3) I still worry about the accuracy of the TRMM percipitation and GRACE data. The daily TRMM precipitation is very poor and the montly TRMM precipitaiton is also not good. The spatial resolution of GRACE is low, and the accuracy maybe not good. 

Response: Yes, you are right. The point-wise comparison is not good. It is very bad in some regions, like Mekong River basin, as we tested. Therefore, we provided a TRMM validation against the observed gauge station data in the Yangtze River Basin in our manuscript. It was shown to have good correlation and RMSE values, as stated in Section 3. Since our main purpose is not a point-wise validation but estimating the runoff of the entire Yangtze River Basin based on remote sensing, the gridded data for the entire basin, in an average statistical sense, should yield an estimated quantity at an acceptable level. Or else, no one will investigate this approach.

4) In China, the runoff is seriously affected by human activities and water management projects, but this paper didn't consider the above factors. So, I think the accuracy of this research is questioned.

Response: To accommodate your request, we added one and a half paragraph and related citations in the introduction of the revised manuscript as below:

" In China, the food security relies on the water resources of the YRB; hence, this river basin is one of the most significant study area [53]. A number of water management projects have been operated for adjustment of R during severe drought and flood seasons [54-55]. Other human activities, such as damming, groundwater withdrawal, water consumption and land use change, can have substantial impacts on R of the river basin [56,57]. Land-use change accounts for < 0.2% of the change in the runoff trend [60], while damming, groundwater withdrawal, and water consumption account for < 2%, < 1%, and < 10% of the changes in the annual runoff [62], respectively. However, other research studies have indicated climate change as the main factor affecting R (accounting for 90% of the change in R in the YRB) [58-61]. Additional studies have indicated that human activities in the YRB might significantly influence changes in the sediments, but not in R [59,63].

Notably, the R values estimated from the water balance equation were calculated by subtraction among the remotely-sensed P, ET, and ∆S: the systematic effect caused by the YRB environment and by human activities should have been partly mitigated by this subtraction process. The aforementioned information partially supports our data-driven R estimation for the YRB,  based purely on the above remotely-sensed hydrological variables on the application of the water balance equation. After calculating the resulting R, we compared our results with those published in Syed et al. (2009) [44] and Ferreira et al. (2013) [37]; moreover, we performed a seasonal accuracy assessment of the observed R time series, in order to evaluate the performance of our method during all seasons. Finally, we discussed the deficiencies of modeled data products and tried to explain the reasons for different seasonal discrepancies between the R obtained through our method and the observed in-situ data."

5) Language should be improved and polished.

Response: To accommodate your request, we have paid for the English editing service before re-submitting the revised manuscript. 

Reviewer 2 Report

Title: An Improved Remotely-Sensed Total Basin Discharge with its Seasonal 

Error Characterization in the Yangtze River Basin

Authors: Yutong Chen, Hok Sum Fok *, Zhongtian Ma, Robert Tenzer

Satellite remote sensing observations play an important role in research and applications. Ground-based observations are limited, especially in remote continents and mountainous regions. The paper uses remote sensing data to improve total basin discharge estimates, which is a new approach to an important research topic. 

Major issues:

Did the authors forget to describe the method to compute the discharge/runoff? The authors talked about the remote sensing data, but no description for the method is given.

TRMM 3B43 is not a 100% satellite-based product. The GPCC gauge data are used for bias correction. Due to its coarse resolution (1 degree), biases can still be seen in the figures.

Where is the rainfall data source? No reference and no URL. Are the data publicly available?

Missing important references for TRMM:

Huffman, G.J. (1997), Estimates of Root-Mean-Square Random Error for Finite Samples of Estimated Precipitation, J. Appl. Meteor.

Huffman, G.J., R.F. Adler, D.T. Bolvin, G. Gu, E.J. Nelkin, K.P. Bowman, Y. Hong, E.F. Stocker, D.B. Wolff (2007), The TRMM Multi-satellite Precipitation Analysis: Quasi- Global, Multi-Year, Combined-Sensor Precipitation Estimates at Fine Scale., J. Hydrometeor

Huffman, G.J., R.F. Adler, D.T. Bolvin, E.J. Nelkin (2010), The TRMM Multi-satellite Precipitation Analysis (TMPA). Cpter 1 in Satellite Rainfall Applications for Surface Hydrology, doi:10.1007/978-90-481-2915-7

https://docserver.gesdisc.eosdis.nasa.gov/public/project/GPM/3B42_3B43_doc_V7.pdf

Liu, Z. D. Ostrenga, W. Teng and S, Kempler, 2012, Tropical Rainfall Measuring Mission (TRMM) Precipitation Data Services for Research and Applications, Bulletin of the American Meteorological Society, doi: http://dx.doi.org/10.1175/BAMS-D-11-00152.1

Dated URL for TRMM data download, mirador => https://disc.gsfc.nasa.gov/datasets/TRMM_3B43_V7/summary

Minor:

Line 14, Abstract: what time period this work is about?

Line 14, Abstract: Might want to mention the remote sensing product names

Line 32, might want to add the remote sensing product names as keywords

Line 142, any reference on this association with the El Nino event?

Line 169, spell out TRMM

Author Response

Please read the attached PDF, as some equations and symbols cannot be displayed. Thank you.

Comments and Suggestions for Authors

Title: An Improved Remotely-Sensed Total Basin Discharge with its Seasonal 

Error Characterization in the Yangtze River Basin

Authors: Yutong Chen, Hok Sum Fok *, Zhongtian Ma, Robert Tenzer

Satellite remote sensing observations play an important role in research and applications. Ground-based observations are limited, especially in remote continents and mountainous regions. The paper uses remote sensing data to improve total basin discharge estimates, which is a new approach to an important research topic.

Major issues:

Did the authors forget to describe the method to compute the discharge/runoff? The authors talked about the remote sensing data, but no description for the method is given.

Response: Yes, you are right. In essence, we mentioned our method (i.e., water balance equation) in the introduction. It appears that all the reviewers have the same question. Therefore, we added in section 3.1 in the revised manuscript as below:

"3. Methodology, Data Description and Validation

3.1 Methodology

The terrestrial water balance equation, based on the principle of mass conservation, has been used to estimate the remotely-sensed runoff R for the YRB [41-42]:

                                                                                                                                                                                             (1)

where P is the precipitation (mm/month), ET the evapotranspiration (mm/month), ∆S the terrestrial water storage change (mm/month), which can be expressed as:

                                                                                                                                            (2)

where  is water storage anomaly (mm/month) of month i.

Table 1 summarizes all the remotely-sensed data products and the in-situ gauge observations used in this study. All the in-situ gauge data time series were obtained from the Changjiang Water Resources Commission (Ministry of Water Resources; http://www.cjh.com.cn). Notably, the remotely-sensed P data from the TRMM, the ET data from the MODIS, and the S data from GRACE have different spatial resolutions (Table 1) and need has to be unified. Therefore, the TRMM and MODIS data were re-sampled at a  resolution, whereas the GRACE data were interpolated at a  resolution. The remotely-sensed data were then validated against the in-situ gauge at point locations. No in-situ ET gauge stations were available in the YRB [52]; hence, we needed to use the evaporation (E) in-situ measurements. Since terrestrial water storage measurements were unavailable at a point location, the GLDAS was employed to validate the data in this study.

Once the steps described above were completed, all the P, ET, and  values gridded within the YRB up to Datong station location were individually summed up and used to calculate the monthly R values between 2003 and 2012 (see equation (1) and Figure 1). To be comparable with the unit of the estimated R values, the daily observed discharge (ec) data of Datong station were combined to in the form of monthly discharge values and subsequently divided by the approximate total drainage area, up to Datong station (i.e., ) to obtain the monthly R per unit area (mm/month) (Figure 1)."

Table 1. Remotely-sensed data products and in-situ gauge observations used in this study.

Hydrological   Component

Data Source

Spatial   Resolution

Temporal   Resolution

Precipitation

TRMM 3B43 v7

Monthly

Gauges

Point

Daily

Evapotranspiration

MOD16A2

Monthly

GLDAS

Monthly

Evaporation

Gauges

Point

Daily

Terrestrial   water storage change

GRACE

Monthly

GLDAS

Monthly

Runoff

Gauges

Point

Daily

TRMM 3B43 is not a 100% satellite-based product. The GPCC gauge data are used for bias correction. Due to its coarse resolution (1 degree), biases can still be seen in the figures.

Response: Yes, you are right. We added a 'note' sentence right after the TRMM data description as below:

"... Notably the TRMM 3B43 V7 data products are not purely a satellite-based; in fact, the Global Precipitation Climatology Centre (GPCC) gauge data were used for bias correction and calibration [76-78]. "

Where is the rainfall data source? No reference and no URL. Are the data publicly available?

Response: The ground-observed precipitation, evaporation, and discharge data are purchased from Changjiang (Yangtze) Water resources commission, Ministry of water resources (http://www.cjh.com.cn) under National Basic Research Program of China (973 program) (Grant No.: 2013CB733302). In other words, they are not publicly available without signing their agreement and contract. We stated it in the original "In situ hydrological gauge stations" in section 3.1 and in the acknowledgement section.

Missing important references for TRMM:

Huffman, G.J. (1997), Estimates of Root-Mean-Square Random Error for Finite Samples of Estimated Precipitation, J. Appl. Meteor.

Huffman, G.J., R.F. Adler, D.T. Bolvin, G. Gu, E.J. Nelkin, K.P. Bowman, Y. Hong, E.F. Stocker, D.B. Wolff (2007), The TRMM Multi-satellite Precipitation Analysis: Quasi- Global, Multi-Year, Combined-Sensor Precipitation Estimates at Fine Scale., J. Hydrometeor

Huffman, G.J., R.F. Adler, D.T. Bolvin, E.J. Nelkin (2010), The TRMM Multi-satellite Precipitation Analysis (TMPA). Chapter 1 in Satellite Rainfall Applications for Surface Hydrology, doi:10.1007/978-90-481-2915-7

https://docserver.gesdisc.eosdis.nasa.gov/public/project/GPM/3B42_3B43_doc_V7.pdf

Liu, Z. D. Ostrenga, W. Teng and S, Kempler, 2012, Tropical Rainfall Measuring Mission (TRMM) Precipitation Data Services for Research and Applications, Bulletin of the American Meteorological Society, doi: http://dx.doi.org/10.1175/BAMS-D-11-00152.1

Response: Yes, we cited them in the revised manuscript accordingly.

Dated URL for TRMM data download, mirador => https://disc.gsfc.nasa.gov/datasets/TRMM_3B43_V7/summary

Response: Done, Thank you.

Minor:

Line 14, Abstract: what time period this work is about?

Response: It is between January 2003 and December 2012. We added it into the abstract.

Line 14, Abstract: Might want to mention the remote sensing product names

Response: Yes, we added it into the abstract as below:

" In this study, we conducted total basin discharge estimation of the Yangtze River Basin, based purely on remotely-sensed data. This estimation considered the period between January 2003 and December 2012 at a monthly temporal scale, and was based on precipitation data collected from the Tropical Rainfall Measuring Mission (TRMM) satellite, evapotranspiration data collected from the Moderate Resolution Imaging Spectroradiometer (MODIS) satellite, and terrestrial water storage data collected from the Gravity Recovery and Climate Experiment (GRACE) satellite."

Line 32, might want to add the remote sensing product names as keywords

Response: Yes, we changed the keywords as: "Total basin discharge; GRACE satellite gravimetry; TRMM satellite precipitation; MODIS satellite-derived evapotranspiration; Yangtze River Basin"

Line 142, any reference on this association with the El Nino event?

Response: A website (https://ggweather.com/enso/oni.htm) described the Oceanic Nino index (ONI) and defined the strength of each ENSO event. Therefore, we changed the sentence as below:

" The exceptionally large R that occurred in July 2010 was likely caused by a moderate El Niño event that started in autumn 2009, as also indicated by the Oceanic Niño Index (https://ggweather.com/enso/oni.htm)."

Line 169, spell out TRMM

Response: We corrected accordingly.

Reviewer 3 Report

In the manuscript the authors compare in situ hydrological gauge station data with the observations obtained by remote sensing techniques, as TRMM, MODIS an GRACE.

Before giving my positive opinion for publishing this paper I need some clarification, as reported below. 

In the definition of the statistic indices PCC. RMSE and NSE are not clear the terms in the formulas (1), (2) and (3): Xo is called the observed value  and Xm is called the estimated value. What do you mean?  Is Xm the measure with the tide gauge and Xo the value obtained by the remote sensing technique? 

It seems that you are using the tide gauge values as model values. 

I think that you should clarify better this point, because, by my point of view both data (tide gauge and remote sensing) are observations.

In all the work, you use three statistical parameters PCC, RMSE and NSE. Could you explain which different information do yo expect by these three parameters? For instance, the NSE is a static index to evaluate the model performance, but what is your model? 

In Table 4, the authors compare the terrestrial water storage obtained by GRACE with the GLDAS model. The NSE index present negative values. Could you comment these results?

Author Response

Please refer to the attached PDF since some equations and symbols cannot be displayed. Thank you.

In the manuscript the authors compare in situ hydrological gauge station data with the observations obtained by remote sensing techniques, as TRMM, MODIS an GRACE.

Before giving my positive opinion for publishing this paper I need some clarification, as reported below. 

In the definition of the statistic indices PCC. RMSE and NSE are not clear the terms in the formulas (1), (2) and (3): Xo is called the observed value  and Xm is called the estimated value. What do you mean?  Is Xm the measure with the tide gauge and Xo the value obtained by the remote sensing technique? 

Response: Thank you for comment. We confess our misusage of NSE in remotely sensed data validation section, but we reserved NSE to compare our estimated runoff based on water balance model against the ground-based runoff observation. The terms in the formulas (1), (2), and (3) are refined for clarification in the below response.

It seems that you are using the tide gauge values as model values. 

I think that you should clarify better this point, because, by my point of view both data (tide gauge and remote sensing) are observations.

Response: Yes, we clarify it in the below.

In all the work, you use three statistical parameters PCC, RMSE and NSE. Could you explain which different information do yo expect by these three parameters? For instance, the NSE is a static index to evaluate the model performance, but what is your model? 

Response: We confess the misusage of NSE in the data validation section. We deleted NSE metrics in the data validation part. All the data used in data validation part are observations. PCC was used to describe the degree of linear correlation between remotely-sensed data and ground-observed data. The closer PCC to 1, the better the fit of remotely-sensed data and ground-observed data are. RMSE was used to measure the deviation between the remotely-sensed data and ground-observed data. The smaller RMSE is, the better the remotely-sensed data is. The NSE is a static index to compare the estimated value from the model against the gauge value. In our case, the water balance model is our model.

In order to clarify the metrics usage, data, and model comparison, the section of "Evaluation Metrics" of the manuscript are revised as below:

"3.3.1 Evaluation metrics

To estimate the remotely-sensed R for the YRB, several remotely-sensed datasets have been considered (i.e., P (TRMM), ET (MODIS), and S (GRACE) data). These datasets and the estimated R were subjected to an accuracy evaluation, by comparing them to the gauge station observed time series. In particular, the remotely-sensed data were compared to the gauge station observed time series using the PCC and the RMSE, whereas the R values estimated from the water balance model were compared to the gauge station observed time series using the PCC, RMSE, and the Nash-Sutcliffe model efficiency (NSE) coefficient. The PCC is a number that represents the strength of the linear relationship between two data time series; it can be calculated as follows:

                                                                                                                                          (1)

The RMSE represents the accuracy of the estimation; It can be calculated as follows:

                                                                                                               (2)

The NSE coefficient proposed by Nash and Sutcliffe (1970) [73] is usually used to assess hydrological models; it can be calculated as follows:

                                                                                                              (3)

where  and  are the gauge observed and the average gauge observed values, respectively;  and irepresent the remotely-sensed R and the average remotely-sensed values, respectively; and N corresponds to the number of observations within the time series. The closer the NSE is to 1, the better the performance of  is. Notably the numerical value of NSE should be equivalent to the coefficient of determination (R2), a statistical measure used to predict future outcomes and test hypotheses."

In Table 4, the authors compare the terrestrial water storage obtained by GRACE with the GLDAS model. The NSE index present negative values. Could you comment these results?

Response: As mentioned above, we have deleted NSE in the data validation part, but reserved NSE to compare our estimated runoff based on water balance model against the ground-based runoff observation. For the explanation of negative values, it indicates the observed mean of GRACE data is a better predictor than GLDAS-modeled data. The explanation can be found in:

https://agrimetsoft.com/calculators/Nash%20Sutcliffe%20model%20Efficiency%20coefficient.aspx

Round 2

Reviewer 1 Report

The authors gave a good response to my previous comments. But there is still some problems:

1) In table 1, the authors should expain why you used the pan evaporation data?  

2) In line 189-190, the auhtors said: "To validate the remotely-sensed P (TRMM) and ET (MODIS) data, we considered the observed P and E time series collected at stations..." 

    But I confused that the ET(MODIS) cannot be validated by the observed E data series.

    because Modis ET is actual ET, but pan evaporation is potential ET.

3) The analysis in Figure 4  is also inexplicable.

4) Part 4 should be discussion, not results and discussion.

5) Rewrite Part 5.

Author Response

Please refer to the attached PDF, since Figure and Table cannot be displayed here.

Comments and Suggestions for Authors

The authors gave a good response to my previous comments. But there is still some problems:

Response: Thank you. To make it easier for reading the revised manuscript of the 2nd round comment, the previous change due to language and others are not marked in red. Only the change made for this 2nd round comment will be marked in red.

1) In table 1, the authors should expain why you used the pan evaporation data?  

Response: It is because no in-situ stations for Evapotranspiration (ET) observations are available in Yangtze River Basin, as shown from Figure 1 of Khan et al. (2018) (reference citation number [52] of the manuscript) as below:

In fact, we stated the reason in line 193-194 of the 1st round revised manuscript as below:

"No in-situ ET gauge stations were available in the YRB [52]; hence, we needed to use the evaporation (E) in-situ measurements."

Reference:

52. Khan, M. S.; Liaqat, U. W.; Baik, J.; Choi, M. Stand-alone uncertainty characterization of GLEAM, GLDAS and MOD16 evapotranspiration products using an extended triple collocation approach. Agr. Forest Meteorol. 2018, 252, 256-268.

2) In line 189-190, the auhtors said: "To validate the remotely-sensed P (TRMM) and ET (MODIS) data, we considered the observed P and E time series collected at stations..." 

    But I confused that the ET(MODIS) cannot be validated by the observed E data series.

    because Modis ET is actual ET, but pan evaporation is potential ET.

Response: Yes, you are right. ET(MODIS) cannot be validated by the observed E data series, since they are not the same quantity. However, as mentioned and shown in the above figure, we did not have actual ET observations available in Yangtze River Basin. In essence, it is not easy to observe actual ET. So, pan evaporation data time series were used as data for validation in the original manuscript.

Based on your comments and in order to make them comparable, we just found a widely-used formula (Zhang et al., 2001) in China for estimating actual ET from pan evaporation. Based on vegetation types shown in Figure 1 of Khan et al. (2018), we choose suitable plant-available water coefficient (w) values for the formula accordingly (i.e., w = 0.5 for short grass and crops; w= 2.0 for forest). Therefore, we calculated actual ET, followed by modifying our Figure 4 and statistics in Table 3.

The sentence in line 189-190 is then modified accordingly as below:

" To validate the remotely-sensed P (TRMM) and ET (MODIS) data, we considered the observed P and the inferred ET using equation (6) of [73] from the observed E data time series collected at stations..."

Reference:

Zhang, L., Dawes, W. R., & Walker, G. R. (2001). Response of mean annual evapotranspiration to vegetation changes at catchment scale. Water resources research, 37(3), 701-708.

3) The analysis in Figure 4  is also inexplicable.

Response: we responded in the above. To be clear, the revised Figure 4 and Table 3 are shown in the below:

Table 3. Statistical comparison between the remotely-sensed MODIS evapotranspiration data and the inferred evapotranspiration from the observed evaporation data at the three selected stations along the upper, middle, and lower reaches of the Yangtze River basin.

Station

PCC

RMSE   (mm/month)

Pingshan

0.87

26.4

Chenglingji

0.71

29.7

Datong

0.80

29.4

Figure 4. MODIS evapotranspiration and inferred evapotranspiration from observed evaporation  data time series in correspondence of (a) Pingshan (upper reaches), (b) Chenglingji (middle reaches), and (c) Datong (lower reaches) stations.

4) Part 4 should be discussion, not results and discussion.

Response: We changed accordingly.

5) Rewrite Part 5.

Response: We partly rewrite part 5 for the above missing analysis part. The rewritten paragraphs in Part 5 are as below:

"5. Conclusions

Previous studies has calculated the time series of monthly TBD (in terms of R) in the YRB applying the water balance equation to a combination of RS and modeled data products. Here, we applied instead a data-driven method purely based on RS data. Before the investigation, the remotely-sensed data were first validated against the in-situ gauge measurements or the inferred measurements at point locations. This validation process indicated that large uncertainties existed in the modeled data products, as verified when the modeled data products were compared with the observed hydrological data collected from the in situ stations or the inferred data, or when the estimated runoff were compared against the observed runoff.

Our best R (obtained from purely remotely-sensed data) and those of Ferreira et al. (2013) [37] against the observed runoff reveal the PCC of 0.89 and 0.74, and the RMSE of 11.69 mm/month and 14.30 mm/month, respectively: our method showed statistically better results. The peak-to-peak correlation values were also calculated: also in this sense, our method produced slightly better result than those of Syed et al. (2009) [44] and Ferreira et al. (2013) [37].

A seasonal error characterization were conducted to assess the performance of our method during specific season. We found that the remotely-sensed TBD did not capture accurately the maximum and minimum runoff values in summer and winter, respectively. This poor performance could be attributed to an overestimation and underestimation of ET, respectively, which depend on the input variables (e.g., vegetation cover types, solar radiation, and temperature) [52]. The ET values tend to be overestimated or underestimated when the input variables are instable, particularly under dry and humid seasons. Notably the low temporal resolution of the data at a monthly scale could have also contributed to these effects. This finding has not been reported in previous studies.

Satellite data products with higher temporal resolution are gradually becoming available (e.g., daily TRMM precipitation [93], 8-day MODIS evapotranspiration [94], and daily GRACE terrestrial water storage data products [95]). Future research might contemplate the application of our proposed method to these new data, while cautions have to be taken with the data validation, data post-processing steps, and the geographic region."

Reviewer 2 Report

I am glad that the authors have answered all the questions I have. The paper is in a good shape and is ready for publication.

Author Response

Thank you.

Reviewer 3 Report

The authors have improved the presentation of their work, following  most of the suggestion of the reviewers of the previous submission of manuscript, so I think that the paper can be accepted in this form.

Author Response

Thank you.